# Neural Implementations of Rational Approximation: CauchyNet and XNet

## Abstract

Rational approximants often outperform polynomials, especially near nonsmooth structure. On bounded 1D domains, they attain optimal rates (exponential for analytic targets; root-exponential for analytic functions with finitely many singularities). Yet scalable neural parameterizations with classical rates are limited. We propose **CauchyNet**, a rational parameterization from the Cauchy integral formula that we implemented in a neural network. For scalability, we employ **XNet**, a ridge-projected Cauchy layer with linear $\mathcal{O}(MN)$ complexity.

Across parameter-matched approximation and PDE tests, Cauchy-based models show the expected rate diagnostics and strong accuracy–compute trade-offs.

## 1 Introduction

Rational functions—ratios of polynomials—are a cornerstone of classical approximation theory. They often achieve exponential or near-exponential convergence rates where polynomials struggle, particularly for functions with singularities or sharp local features (Newman, 1964; Stahl, 2003). Despite this theoretical power, their adoption within deep learning has been limited. The primary obstacle has been the absence of a framework that is both **scalable** to high dimensions and **grounded in classical theory**, leaving a significant gap between theoretical potential and practical application.

Existing approaches fall short on either scalability or theoretical guarantees. On one hand, classical algorithms such as AAA (Nakatsukasa et al., 2018) and AAA–Lawson variants (Driscoll et al., 2024; Zhang & Han, 2024) achieve near-optimal performance but are computationally intensive and do not readily scale to the high-dimensional settings typical of modern machine learning (Boyd, 2001; Trefethen, 2019). On the other hand, recent neural networks with rational-like activation functions have shown empirical promise (Molina et al., 2020; Boulle et al., 2020), but they often lack the formal connection to classical theory needed to guarantee the celebrated convergence rates.

This leaves a critical question unanswered: can we design a neural architecture that truly embodies the power of rational approximation in a scalable way?

In this work, we provide an affirmative answer by systematically bridging this gap. We develop two complementary Cauchy-inspired neural modules: one that directly realizes classical theory, applicable to low dimensional implementations, and the other one through affine projections and activation functions, making it practical and scalable, hence applicable to high dimensional scenarios.

First, we introduce **CauchyNet**, a novel architecture that provides a direct, theoretically-pure parameterization of multivariate partial fractions via the multidimensional Cauchy formula:

$$\hat{r}_M(x) = \sum_{m=1}^{M} c_m \prod_{j=1}^{N} \frac{1}{\xi_j^{(m)} - x_j}.$$

This construction ensures that CauchyNet represents a genuine class of rational functions, allowing us to prove that it *inherits* the classical exponential and root-exponential convergence rates. While theoretically fundamental, its tensor-product nature presents inherent challenges for scaling to high dimensions.

To overcome this limitation, we adopt **XNet** (Li et al., 2025), a practical and scalable architecture that achieves linear $O(MN)$ complexity. Instead of a direct multivariate construction, XNet applies

a one-dimensional Cauchy activation (introduced in Li et al. (2025)) to learned affine projections (ridges):

$$\hat{r}_M(x) = \sum_{k=1}^{M} a_k \frac{\ell_{1k}(w_k^\top x + b_k) + \ell_{2k}}{(w_k^\top x + b_k)^2 + d_k^2}.$$

This design offers the best of both worlds. We prove that under a mild ridge-separability condition, XNet *preserves* the powerful approximation rates established for CauchyNet, thus successfully translating theoretical power into a practical tool.

**Contributions.**

- **Theory.** We formalize CauchyNet as the native multivariate partial–fraction layer and prove rate inheritance under standard analyticity and pole–separation assumptions; for XNet we prove rate preservation in the ridge–separable regime.
- **Scalability.** XNet provides linear $O(MN)$ complexity while retaining the above guarantees under implementable conditions.
- **Evidence.** Parameter–matched studies on approximation and PDE testbeds show stable training and favorable error–compute Pareto over smooth–activation baselines; XNet also functions as a drop–in flow layer without losing exact tractability.

This work offers the first framework to successfully connect the power of classical rational approximation theory with a scalable and effective architecture for modern deep learning. Section 2 details the derivation of both architectures, while Section 3 presents the comprehensive experimental validation.

## 2 FROM CAUCHY INTEGRALS TO SCALABLE NEURAL ARCHITECTURES

**Multivariate Cauchy on product domains (setup).** Let $U = \prod_{j=1}^{N} U_j \subset \mathbb{C}^N$ be a product domain with each $U_j \subset \mathbb{C}$ simply connected with piecewise smooth boundary, and let $f$ be holomorphic on an open neighborhood of $\overline{U}$. Writing $P = \partial U_1 \times \cdots \times \partial U_N$, the product Cauchy formula yields

$$f(z_1, \ldots, z_N) = \left(\frac{1}{2\pi i}\right)^N \int \cdots \int_P \frac{f(\xi_1, \ldots, \xi_N)}{\prod_{j=1}^{N}(\xi_j - z_j)} \, d\xi_1 \cdots d\xi_N, \qquad (z_1, \ldots, z_N) \in U, \quad (1)$$

obtained by iterating the 1D Cauchy formula in each coordinate.

*Remark* 1 (Interpretation). Equation equation 1 expresses pointwise evaluation of $f$ from boundary values via a kernel with simple poles. This is the analytic template our architectures discretize: after quadrature, equation 1 becomes a finite linear combination of multivariate partial fractions.

We derive CauchyNet from the Cauchy integral formula and its scalable counterpart XNet, then analyze rates and complexity.

### 2.1 CAUCHYNET: ONE-DIMENSIONAL PARTIAL FRACTIONS

We begin with the one-dimensional Cauchy formula. For a simply connected domain $U \subset \mathbb{C}$ with piecewise smooth boundary and $z \in U$,

$$f(z) = \frac{1}{2\pi i} \int_{\partial U} \frac{f(\xi)}{\xi - z} \, d\xi. \tag{2}$$

Discretizing equation 2 with quadrature nodes $\{(\xi^{(m)}, w_m)\}_{m=1}^{M}$ on $\partial U$ yields the rational ansatz

$$\hat{r}_M(x) = \sum_{m=1}^{M} \frac{c_m}{\xi^{(m)} - x}, \qquad c_m = \frac{1}{2\pi i} \, w_m \, f(\xi^{(m)}), \quad x \in K \subset \mathbb{R}. \tag{3}$$

Let

$$\mathcal{H}_{\text{Cauchy}}^{(1D)} = \text{span}\{ (\xi - x)^{-1} : \xi \in \mathbb{C}, \, \text{dist}(\xi, K) \geq \delta \}$$

denote the span of simple 1D Cauchy atoms. One can show that the span can approximate an analytic function with exponential accuracy with respect to the number of atoms used. Functions with finite smoothness can be approximated with polynomial rate.

**Relation to rational approximation.** If $R = P/Q$ is a proper rational function ($\deg P < \deg Q$) and all *denominator zeros* are simple, standard partial fractions expand $R$ into $\{(\xi - x)^{-1}\}$, hence $R \in \mathcal{H}_{\text{Cauchy}}^{(1D)}$. Writing $Q(z) = \prod_{j=1}^{J}(z - \xi_j)$ with distinct simple zeros $\{\xi_j\}$, the coefficients admit the explicit residue formula

$$\frac{P(z)}{Q(z)} = \sum_{j=1}^{J} \frac{\alpha_j}{z - \xi_j}, \quad \alpha_j = \frac{P(\xi_j)}{Q'(\xi_j)}. \tag{4}$$

For general cases, there are two obstructions: (i) *improperness* ($\deg P \geq \deg Q$), which produces a nontrivial polynomial part via long division; and (ii) *multiple/degenerate denominator zeros*, which force higher-order terms $(x - \xi)^{-k}$ ($k \geq 2$) in partial fractions, not contained in the simple-atom span. However, multiple zeros can be avoided via a small perturbation. Likewise, an improper polynomial can be made proper by adding a small high order polynomial to the denominator.

**Inheritance of classical 1D rates.** Classical 1D rational approximation on compact real sets (e.g., Newman (1964); Stahl (2003)) provides near-best $R_M$ with denominator zeros $\delta$-separated from $K$ and

$$\|f - R_M\|_{L^\infty(K)} \leq \begin{cases} C_1(\delta)\,\rho(\delta)^{-M}, & \text{analytic across a } \delta\text{-complex neighborhood,} \\ C_2(\delta)\,\exp(-c(\delta)\sqrt{M}), & \text{analytic off finitely many singularities.} \end{cases}$$

It follows from the above partial fractions analysis that *CauchyNet (1D) inherits the exponential/root–exponential rates* with constants depending only on the fixed separation margin $\delta$.

**Inheritance of High-Dimensional Rates.** The high-dimensional CauchyNet, a direct tensor-product construction from the multivariate Cauchy formula equation 1, parameterizes a genuine class of multivariate partial fractions. It therefore directly inherits the classical approximation rates on bounded domains: exponential for analytic targets and root-exponential for functions with finite singularities, contingent on a fixed pole-separation margin.

**Scalability considerations.** A direct tensorization of equation 2 to $N$ variables produces product atoms $\prod_{j=1}^{N}(\xi_j - x_j)^{-1}$ and, under product quadrature with $m_j$ nodes per coordinate, a term count $M = \prod_{j=1}^{N} m_j$ (exponential in $N$). For low dimensional applications, such as Navier-Stokes equation and other low dimensional PDEs, CauchyNet provides a native adaptation of rational approximation in neural network.

## 2.2 XNET: RIDGE-PROJECTED 1D CAUCHY ATOMS (LINEAR COST)

We use a standard affine map followed by a one-dimensional Cauchy activation per unit, matching the XNet construction of Li et al. (Li et al., 2025). Given $x \in \mathbb{R}^N$, each hidden unit applies

$$\phi_{\text{Cauchy}}(t; \ell_1, \ell_2, d) = \frac{\ell_1 t + \ell_2}{t^2 + d^2}, \quad t = w^\top x + b, \; d > 0, \tag{5}$$

and the network outputs

$$\hat{r}_M(x) = \sum_{k=1}^{M} a_k \, \phi_{\text{Cauchy}}(w_k^\top x + b_k; \ell_{1k}, \ell_{2k}, d_k). \tag{6}$$

We train $(W, b, a)$ and per-unit $(\ell_{1k}, \ell_{2k}, d_k)$, with $d_k = \text{softplus}(\hat{d}_k) + d_{\min}$ for stability. This yields $\mathcal{O}(MN)$ parameters/FLOPs and coincides with the 1D CauchyNet when $N = 1$ (up to reparameterization).

**Two high-dimensional extensions.** *(A) High-dimensional CauchyNet (naive tensor product).* Define the product span

$$\hat{r}_M(x) = \sum_{m=1}^{M} c_m \prod_{j=1}^{N} \frac{1}{\xi_j^{(m)} - x_j}, \qquad x \in K \subset \mathbb{R}^N, \tag{7}$$

with complex *denominator zeros* $\xi^{(m)} \in \mathbb{C}^N$ separated from $K$. While evaluation is $O(MN)$, $M$ typically scales as $\prod_{j=1}^{N} m_j$.

*(B) XNet (ridge-projected).* Our scalable extension is to retain the 1D atom along learned directions, i.e., use equation 6 with $w_k \in \mathbb{R}^N$. This class is distinct from equation 7: it avoids the exponential tensor size and targets affine-denominator effects along learned ridges while keeping linear complexity. For $N=1$, the two classes coincide up to a reparameterization.

**How the 1D rates lift to higher dimensions.** *Analytic targets.* On product domains, the tensor–product discretization of the multivariate Cauchy formula (§2) yields separable partial fractions; the approximation error decays exponentially in the per–axis quadrature order (hence super–polynomial in the total number of atoms), so CauchyNet inherits the 1D exponential behavior (with an exponential term count in $N$). For XNet, the exponential rate transfers under ridge structure (e.g., $f(x) = g(u^\top x)$ or a finite sum of such ridges) via reduction to the 1D setting; absent such structure we refrain from general high–$N$ rate claims.

*Targets analytic off finitely many singularities.* The root–exponential rate carries over via a ridge reduction using XNet: if $f(x) = g(u^\top x)$ (or a finite ridge sum), a ridge-projected Cauchy layer aligned with the ridge direction(s) reduces the problem to 1D and inherits the $\exp(-c\sqrt{M})$ slope (constants depend on separation margins). Outside ridge-structured settings we make expressivity—not rate—claims.

Evidence is empirical (Sec. 3).

## 3 BENCHMARKS: FROM RATE DIAGNOSTICS TO ILLUSTRATIVE CASE STUDIES

**Objective, scope, and protocol.** We evaluate rate predictions under controlled conditions—fixed grids, matched budgets/parameters, and identical optimizers/schedules—using rate readouts in semilog-$M$ (analytic) and log-error vs. $\sqrt{M}$ (finite singularities). Parameter parity is within $0.5\%$ and depth $L$ is held fixed; we stabilize conditioning via $d = \mathrm{softplus}(\hat{d}) + d_{\min}$ with a light margin regularizer (no change to the hypothesis class). Large-scale vision/language results are *illustrative case studies* only and are not used to substantiate exponential or root–exponential claims.

**Choice of PDE diagnostics.** We use three canonical testbeds under strictly matched settings:

- **Burgers (inviscid Riemann):** non-smooth target with a moving shock; Cauchy activation attains earlier and more accurate shock localization.

- **Focusing NLS:** analytic windows on fixed domains; under identical grids/budgets, Cauchy variants achieve lower PDE residuals and state errors with fewer parameters.

- **Kovasznay (steady Navier–Stokes):** closed-form reference enabling direct error evaluation; identical sampling and budgets yield consistently lower solution and residual errors.

**Summary.** Across tasks we observe monotone error decay with capacity and consistent accuracy–compute improvements at fixed budgets; we refrain from attributing precise asymptotic slopes beyond what finite-budget diagnostics support.

### 3.1 FIRST EXPERIMENT: SAME DEPTH ($L=1$ VS. $L=1$)

**Tasks.** Runge $f(x) = \frac{1}{1+25x^2}$ on $[-1, 1]$ (analytic; nearest complex singularities at $\pm 0.2i$) and *Abs-$\alpha$*: $f(x) = |x|^\alpha$ with $\alpha = 0.5$ (root-type singularity at $x = 0$). Theory predicts (near-)exponential accuracy for analytic targets and subexponential/root-exponential otherwise (Newman, 1964; Stahl, 2003).

**Models (like-for-like, $L=1$).** **XNet-1** uses a single hidden layer with an elementwise Cauchy activation function $\phi(z) = \frac{\ell_1 z + \ell_2}{z^2 + d^2 + \varepsilon^2}$, with trainable $\ell_1, \ell_2, d$ and a small $\varepsilon^2$ for numerical stability.

**MLP-1** is a one-hidden-layer fully connected baseline with *tanh*, *GELU*, or *PAU* activations (PAU is itself a rational activation).

**Protocol (parameter/computation matched).** For anchors $w \in \{16, 32, 64, 128\}$ we match the trainable parameter counts across XNet-1 and MLP-1 (mismatch $< 0.5\%$). All models use AdamW, identical training/validation grids, and equal iteration budgets; we report best-validation $\ell_2$.

Table 1: **Validation $\ell_2$ (best checkpoint) for $L{=}1$ vs. $L{=}1$ (seed$= 0$).** Same-parameter, same-depth comparison between **XNet-1** and **MLP-1** baselines (*tanh/GELU/PAU*). *Note:* In 1D, XNet-1 is identical to CauchyNet-1.

| Anchor width (params) | XNet-1 | MLP-1 (tanh) | MLP-1 (GELU) | MLP-1 (PAU) |
|---|---|---|---|---|
| *Runge (analytic)* | | | | |
| 16 ($\sim 113$) | $5.441 \times 10^{-5}$ | $1.752 \times 10^{-2}$ | $2.601 \times 10^{-2}$ | $2.083 \times 10^{-3}$ |
| 32 ($\sim 225$) | $2.233 \times 10^{-5}$ | $1.541 \times 10^{-2}$ | $3.843 \times 10^{-2}$ | $8.693 \times 10^{-4}$ |
| 64 ($\sim 449$) | $2.074 \times 10^{-5}$ | $1.699 \times 10^{-2}$ | $3.736 \times 10^{-2}$ | $4.870 \times 10^{-4}$ |
| 128 ($\sim 897$) | $\mathbf{8.973 \times 10^{-7}}$ | $8.801 \times 10^{-3}$ | $3.954 \times 10^{-2}$ | $7.875 \times 10^{-4}$ |
| $\lvert x \rvert^{0.5}$ *(non-analytic)* | | | | |
| 16 ($\sim 113$) | $1.459 \times 10^{-4}$ | $1.523 \times 10^{-3}$ | $1.331 \times 10^{-3}$ | $1.323 \times 10^{-3}$ |
| 32 ($\sim 225$) | $9.549 \times 10^{-5}$ | $2.102 \times 10^{-3}$ | $1.292 \times 10^{-3}$ | $6.084 \times 10^{-2}$ |
| 64 ($\sim 449$) | $1.858 \times 10^{-4}$ | $2.022 \times 10^{-3}$ | $1.312 \times 10^{-3}$ | $4.963 \times 10^{-4}$ |
| 128 ($\sim 897$) | $9.923 \times 10^{-5}$ | $1.961 \times 10^{-3}$ | $1.693 \times 10^{-3}$ | $2.111 \times 10^{-3}$ |

*Notes.* Parameter counts are matched across models for each anchor width $w$.

**Results (same depth, same params, different rates).** Across matched-parameter, same-depth ($L{=}1$) settings, XNet-1 consistently attains lower errors. On Runge (analytic), semilog-M curves are nearly linear and reach $10^{-5} - 10^{-6}$, while the strongest MLP baseline stays around $10^{-3}$. On $x^{0.5}$ (non-analytic), decay is slower overall but XNet-1 maintains a stable margin (often about one order of magnitude). We therefore report clear accuracy–compute gains and monotone rate diagnostics, without over-interpreting the exact asymptotic slope.

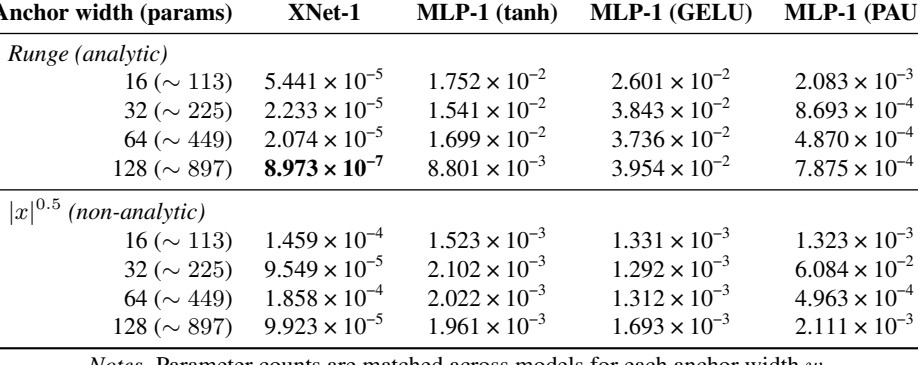

(a) Runge: semilog-$M$    (b) Runge: error vs. params    (c) $\lvert x \rvert^{0.5}$: error vs. params    (d) Training overlays

Figure 1: $L{=}1$ **parameter-matched diagnostics.** (a) Runge shows near-linear semilog-$M$ decay (near-exponential). (b,c) Error vs. parameter count under strict matching. (d) Training overlays.

See Appx. Fig. 4 for the full rate diagnostics (semilog-$M$ for Runge; log-error vs. $\sqrt{M}$ for $\lvert x \rvert^{0.5}$).

*Parameter matching.* Only XNet exposes the atom parameters $(\ell_1, \ell_2, d)$ in addition to width; MLP baselines match total parameters by width ($< 0.5\%$ mismatch).

**Main Findings.** (i) *Analytic $\Rightarrow$ near-exponential.* XNet-1 reaches $10^{-5}$–$10^{-6}$ on Runge where MLP-1 plateaus near $10^{-3}$.

(ii) *Non-analytic $\Rightarrow$ subexponential, still better.* On $\lvert x \rvert^{0.5}$, XNet-1 is uniformly more accurate often an order of magnitude.

(iii) *Same depth.* The advantage holds in a strict $L{=}1$ vs. $L{=}1$ setting—rational, locally supported features deliver strong approximation without extra depth.

## 3.2 BURGERS RIEMANN PROBLEM (1D, INVISCID TARGET)

**Setup.** We adopt the standard inviscid Burgers Riemann configuration on $x \in [-1, 1]$, $t \in [0, 0.6]$ with $(u_L, u_R) = (1, 0)$. The entropy solution features a single moving shock at $x_s(t) = \frac{1}{2}t$, hence the solution is *non-smooth*. This benchmark directly tests a model's ability to handle discontinuities. Our **primary validation metric** is the non-backpropagating shock error $|x_s^{\text{pred}} - x_s^{\text{true}}|$ at $t \in \{0.2, 0.4, 0.6\}$. (Setup follows common practice in the LSNN literature, see Cai et al. (2023); here we isolate the effect of the activation rather than propose a new PDE solver.)

Ground-truth snapshots used for the shock-error computation are shown in Appx. Fig. 5.

We target the inviscid entropy solution; during training we employ a small viscous regularization ($\nu_{\text{train}} = 10^{-3}$) purely for optimization stability, optionally annealed to 0 in the last 30% of steps. All evaluations (Table 2) and ground-truth references use the inviscid PDE ($\nu = 0$).

**Models and protocol.** We use a strong-form PINN with uniform $(x, t)$ sampling and minimize the MSE of a *viscous* conservative residual

$$R_\nu(u) = u_t + \partial_x\left(\tfrac{1}{2}u^2\right) - \nu_{\text{train}}\, u_{xx},$$

with a small $\nu_{\text{train}} = 10^{-3}$ used purely as an optimization stabilizer and *optionally* cosine-annealed to 0 over the last 30% of training steps (setting $\nu_{\text{train}}{=}0$ recovers the inviscid residual).

**XNet**: one hidden layer (width 20) with an elementwise Cauchy activation $\phi(z) = \dfrac{\ell_1 z + \ell_2}{z^2 + d^2}$, trainable $(\ell_1, \ell_2, d)$ and a softplus reparameterization for $d$.

**Baselines**: one-hidden-layer *tanh*/*GELU* at width 35 to match total parameters ($\approx 141$ trainables for all). Samplers, optimizer, and budget are identical across models.

**Metric.** We report the non-backpropagating shock-location error

$$\Delta x_s(t) = \left|x_s^{\text{pred}}(t) - x_s^{\text{true}}(t)\right|, \quad t \in \{0.2, 0.4, 0.6\}, \quad x_s^{\text{true}}(t) = \tfrac{1}{2}t.$$

Slice-wise relative $L^2$ errors (at the same three times) are provided in Appx. A as a *secondary* metric for completeness.

**Results.** The Cauchy-activated PINN attains the lowest shock-location errors and locks onto the discontinuity earlier than smooth-activation baselines under matched parameters.

| Method | Params | $t{=}0.2$ | $t{=}0.4$ | $t{=}0.6$ |
|---|---|---|---|---|
| **PINN + Cauchy (w=20)** | 141 | $3.109{\times}10^{-3}$ | $1.443{\times}10^{-3}$ | $7.303{\times}10^{-4}$ |
| PINN + GELU (w=35) | 141 | $8.785{\times}10^{-3}$ | $2.957{\times}10^{-2}$ | $4.546{\times}10^{-2}$ |
| PINN + tanh (w=35) | 141 | $5.644{\times}10^{-2}$ | $7.270{\times}10^{-2}$ | $9.566{\times}10^{-2}$ |

Table 2: **Shock-location error** $|x_s^{\text{pred}} - x_s^{\text{true}}|$ for the inviscid Riemann problem. Default training uses a small viscosity $\nu_{\text{train}} = 10^{-3}$ (optionally cosine-annealed to 0); evaluations use the inviscid PDE. Full training curves and multi-seed summaries are in Appx. A.

**Remarks on comparability.** LSNN commonly reports slice-wise relative $L^2$ near $10^{-2}$ under a weak-form/discretized-flux training regime with time blocking. Our setup is a strong-form PINN with uniform $(x, t)$ sampling and no shock-focused mechanisms; absolute $L^2$ figures are therefore not strictly comparable, and we emphasize shock-location error under matched parameter budgets. For transparency, preliminary runs with a strictly inviscid residual ($\nu_{\text{train}}{=}0$) yield *similar shock-location errors* but do not reach LSNN's $\sim 10^{-2}$ slice-wise $L^2$; closing this gap likely requires shock-aware sampling or weak-form losses, which is orthogonal to our activation study and left for future work.

## 3.3 Focusing Schrödinger (complex-valued)

We solve the focusing nonlinear Schrödinger equation

$$i\,\psi_t + \tfrac{\sigma}{2}\,\psi_{xx} + \kappa\,|\psi|^2\psi = 0, \qquad x \in [-5,5],\ t \in [0,T],$$

with periodic boundary conditions $\psi(-5,t) = \psi(5,t)$ and $\psi_x(-5,t) = \psi_x(5,t)$. Writing $\psi = u + i\,v$ and splitting into real and imaginary parts gives

$$\begin{cases} u_t + \tfrac{\sigma}{2}\,v_{xx} + \kappa\,(u^2+v^2)\,v = 0, \\ -\,v_t + \tfrac{\sigma}{2}\,u_{xx} + \kappa\,(u^2+v^2)\,u = 0, \end{cases}$$

and we set $\sigma = 1,\ \kappa = 1$.

**Training protocol.** Strong-form residual, double precision; uniform resampling each epoch; Adam for 10,000 epochs at learning rate $10^{-3}$. Sample counts follow the code: collocation $n_c$=20000, initial $n_i$=2048, boundary $n_b$=200. We compare a PINN (4×100 `tanh`, 30,802 params) against a CauchyNet (basis size $m$=400, 2400 params).

Table 3: **Focusing NLS (periodic, strong-form, double precision).** Same grids/optimizer/budget; 10k epochs, lr = $10^{-3}$. CauchyNet uses ~12.8× fewer parameters yet dominates all metrics.

| Metric | PINN (4×100) | CauchyNet ($m$=400) |
|---|---|---|
| Parameters | 30.8k | 2.4k |
| PDE residual (RMS) | $3.571\times10^{-2}$ | $\mathbf{1.538\times10^{-3}}$ |
| State MSE (real) | $2.748\times10^{-3}$ | $\mathbf{5.522\times10^{-7}}$ |
| State MSE (imag) | $6.764\times10^{-3}$ | $\mathbf{1.189\times10^{-6}}$ |
| Rel. error (real) | $1.087\times10^{-1}$ | $\mathbf{1.570\times10^{-3}}$ |
| Rel. error (imag) | $1.940\times10^{-1}$ | $\mathbf{2.528\times10^{-3}}$ |
| BC violation | $8.145\times10^{-6}$ | $\mathbf{3.185\times10^{-8}}$ |

**Main Finding.** With $\sim 12.8\times$ fewer parameters (2.4k vs. 30.8k), CauchyNet achieves $\approx 23\times$ lower PDE residual and $10^3$–$10^4\times$ lower state MSE, indicating the gains come from the rational inductive bias rather than capacity.

## 3.4 Steady Navier–Stokes: Kovasznay flow

**Setup.** We consider the steady incompressible Navier–Stokes equations on $\Omega = [-0.5,1] \times [-0.5,1.5]$ with viscosity $\nu = 0.025$, zero body force $f \equiv 0$, and Dirichlet data taken from the exact solution:

$$-\nu\Delta u + (u\cdot\nabla)u + \nabla p = 0, \qquad \nabla\cdot u = 0 \text{ in } \Omega, \quad u|_{\partial\Omega} = u_\star.$$

The Kovasznay flow (Dockhorn, 2019) provides the closed form (set $\lambda = \tfrac{1}{2\nu} - \sqrt{\tfrac{1}{4\nu^2} + 4\pi^2}$):

$$u_\star(x) = \begin{bmatrix} 1 - e^{\lambda x_1}\cos(2\pi x_2) \\ \tfrac{\lambda}{2\pi}\,e^{\lambda x_1}\sin(2\pi x_2) \end{bmatrix}, \qquad p_\star(x) = \tfrac{1}{2}\big(1 - e^{2\lambda x_1}\big) - \bar{p},$$

where $\bar{p}$ sets the reference level (we use mean-zero pressure on $\Omega$).

**Training protocol.** Strong-form residuals for momentum, divergence and boundary terms (double precision). Adam for 1,000 epochs at learning rate $10^{-3}$; identical collocation/BC sampling across methods. We compare a PINN (4×100 `tanh`, 30,802 params) against a CauchyNet (basis size $m$=100, 600 params).

| Activation | best NLL ↓ | key hparams |
|---|---|---|
| **Cauchy** | **-1.2297** | steps=10k, lr=5e-4, treg=3e-4, sclip=1.5, m=64, width=128 |
| SiLU | -1.1263 | steps=10k, lr=5e-4, treg=2e-4, sclip=1.8, width=180 |
| tanh | -0.4084 | steps=10k, lr=1e-3, treg=1e-4, sclip=2.0, width=180 |
| ReLU | -0.2998 | steps=10k, lr=5e-4, treg=1e-4, sclip=2.0, width=180 |

Table 5: RealNVP on 8G2D ($K$=6, $L$=2, seed= 0). All runs are **parameter-matched** ($\approx$199k trainables). Cauchy improves NLL by 0.10–0.93 nats over smooth activations under the same budget.

Table 4: **Kovasznay flow (steady Navier–Stokes).** Same grids/optimizer/budget; 1k epochs, lr $= 10^{-3}$. CauchyNet uses $\sim$51$\times$ fewer parameters and improves all metrics. *Visual fields and error maps:* see Appx. Fig. 6.

| Metric | PINN ($4\times100$) | CauchyNet ($m$=100) |
|---|---|---|
| Parameters | 30.8k | 0.6k |
| PDE residual (RMS) | $1.210\times10^{-2}$ | $\mathbf{4.769\times10^{-3}}$ |
| $u_1$ MSE | $9.222\times10^{-4}$ | $\mathbf{1.072\times10^{-6}}$ |
| $u_2$ MSE | $2.089\times10^{-4}$ | $\mathbf{7.879\times10^{-7}}$ |
| Rel. err ($u_1$) | $2.414\times10^{-2}$ | $\mathbf{2.603\times10^{-4}}$ |
| Rel. err ($u_2$) | $1.457\times10^{-2}$ | $\mathbf{8.951\times10^{-5}}$ |
| BC violation | $1.506\times10^{-4}$ | $\mathbf{8.127\times10^{-7}}$ |

**Main Finding.** Cauchy-type networks stably approximate coupled nonlinear PDEs and reach $10^2$–$10^3\times$ lower solution errors than standard PINNs under substantially smaller parameter budgets.

### 3.5 GENERATIVE MODELING (TOY FLOW, 8G2D)

We use a RealNVP-style flow with $K$=6 coupling blocks; each $(s,t)$-net is an $L$=2-layer MLP. We compare **Cauchy** against **SiLU**, **tanh**, and **ReLU** under a strictly *parameter-matched* budget ($\approx$199k trainables; seed= 0), and report best validation NLL (lower is better).

**Why Cauchy** Its feature layer is a rational, heavy-tailed, locally supported basis: for each output channel we linearly combine $m$ multivariate *Cauchy activations*, which induces a strong bias for multi-modal ring-like densities and sharp gaps. To match parameters we set Cauchy *width*= 128, *# atoms $m$*=64 and use slightly larger widths ($\approx$ 180) for the other activations; training steps, number of blocks/layers, and dataset are identical across runs. For numerical stability only, we use scale clipping $s \in [-\text{sclip}, \text{sclip}]$ and a tiny $L^2$ on $t$.



Figure 2: **RealNVP on 8G2D (parameter-matched, $K$=6, $L$=2).** Cauchy recovers all eight modes with cleaner gaps and little spurious central mass, consistent with its rational, heavy-tailed inductive bias.

*Architecture sensitivity (keeping $L$=1).* We additionally evaluate a shallower per-block setting with *more coupling blocks but $L$=1* (i.e., one hidden layer inside each $(s,t)$-net). With $(K$=8, $L$=1), Cauchy improves further: **-1.5623** NLL at $\sim$200k params and **-1.4910** at 150k params, outperforming the $(K$=6, $L$=2) setting (Table 6). Figure 3 *shows the two $L$=1 cases side-by-side* (150k vs. 200k), confirming cleaner mode recovery and gaps.

| Config | Width $W$ | partial fraction $m$ | Params | best NLL $\downarrow$ |
|---|---|---|---|---|
| Cauchy ($K{=}6, L{=}2$) | 128 | 64 | $\approx$199k | -1.2297 |
| Cauchy ($K{=}8, L{=}1$) | 96 | 96 | 150,544 | -1.4910 |
| Cauchy ($K{=}8, L{=}1$) | 128 | 96 | 200,208 | **-1.5623** |

Table 6: 8G2D: Cauchy under alternative $(K, L)$ while keeping widths/atom counts modest. More coupling blocks with shallower $(s, t)$-nets ($L{=}1$) yields better NLL at similar or fewer parameters.

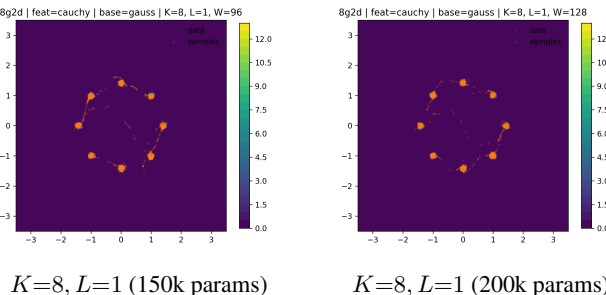

$K{=}8, L{=}1$ (150k params) $\qquad\qquad$ $K{=}8, L{=}1$ (200k params)

Figure 3: **Cauchy architecture sensitivity ($L{=}1$ kept).** Two $L{=}1$ cases (150k vs. 200k) show improved mode recovery and cleaner gaps as coupling stages increase or capacity slightly grows.

**Main Finding.** Under strictly matched parameter budgets, Cauchy layers provide a favorable likelihood–compute trade-off on multi-modal densities. Keeping $L{=}1$ inside $(s, t)$-nets while increasing coupling stages ($K$) further amplifies the gains.

## 4 FUTURE DIRECTIONS

We aim to carry XNet (the Cauchy-activated ridge layer) into foundation-scale settings while keeping theory in the loop:

1. **Scaling to foundation models.** Use XNet as a *drop-in MLP replacement* inside ViT/Transformer blocks under matched FLOPs/memory, and report *loss–compute Pareto* and *long-horizon stability*.

2. **Scientific ML at production scale.** Deploy on turbulence, nowcasting, and reconstruction pipelines with physics-aware metrics (spectral/energy errors, conservation, rollout stability) to assess accuracy–compute trade-offs beyond toy benchmarks.

3. **Systems support for rational layers.** Provide fused kernels and mixed-precision paths, plus margin-aware initialization/regularizers to maintain denominator–set separation $\delta$ at scale; add runtime monitors for conditioning and numerical safety.

4. **Theory at scale: scaling laws.** Develop empirical/theoretical scaling laws linking approximation/risk to width $M$, projection conditioning, and separation $\delta$ under stochastic optimization, extending exponential/root–exponential guarantees toward task-level performance.

ETHICS STATEMENT

This work is foundational, focusing on the theory of rational function approximation for neural networks, and we foresee no direct negative societal impact. Like any machine learning model, downstream applications of our architectures could reflect biases present in training data; mitigating these application-specific risks is the responsibility of the developers. Notably, our methods are designed for computational efficiency, and all experiments were conducted with a minimal environmental footprint.

REPRODUCIBILITY STATEMENT

All experiments are fully reproducible. The code to replicate our results will be made publicly available upon acceptance. All datasets used are standard public benchmarks (e.g., Runge function) or are generated from canonical problems in scientific computing (e.g., Burgers' equation). **Section 3** provides a complete description of the experimental setup, including the strict parameter-matching protocol, model configurations, and all hyperparameters required for replication.

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

# A ADDITIONAL FIGURES

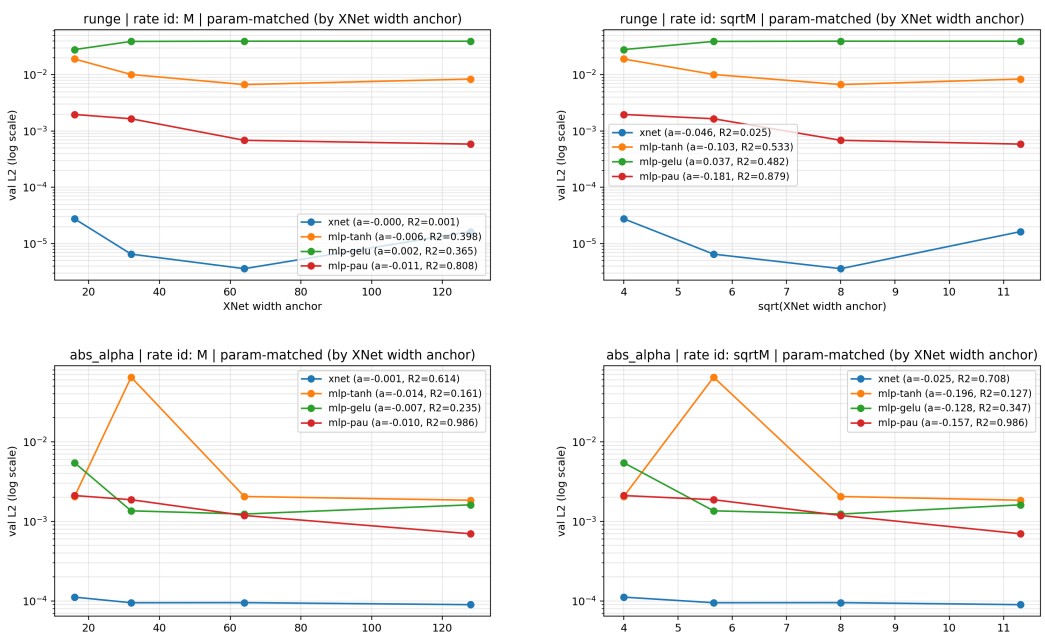

Figure 4: **Experiment 1** (Sec. 3.1): Runge and $|x|^{0.5}$ diagnostics under $L=1$ and parameter-matched settings. Top: Runge shows near-linear semilog-$M$ decay (near-exponential) and its $\sqrt{M}$ view. Bottom: $|x|^{0.5}$ exhibits slower decay in both $M$ and $\sqrt{M}$ views. Plots correspond to the summary in Table 1.

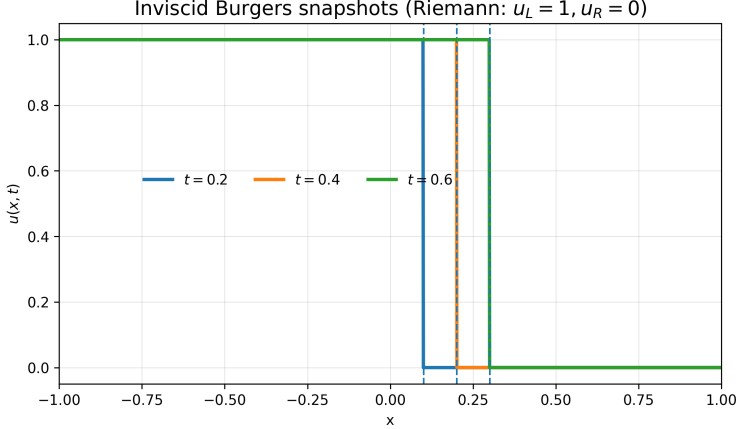

Figure 5: **Burgers (Sec. 3.2).** Ground-truth snapshots at $t \in \{0.2, 0.4, 0.6\}$ with shock $x_s(t) = \frac{1}{2}t$ (dashed). This reference is used to compute the shock-location error in Table 2.

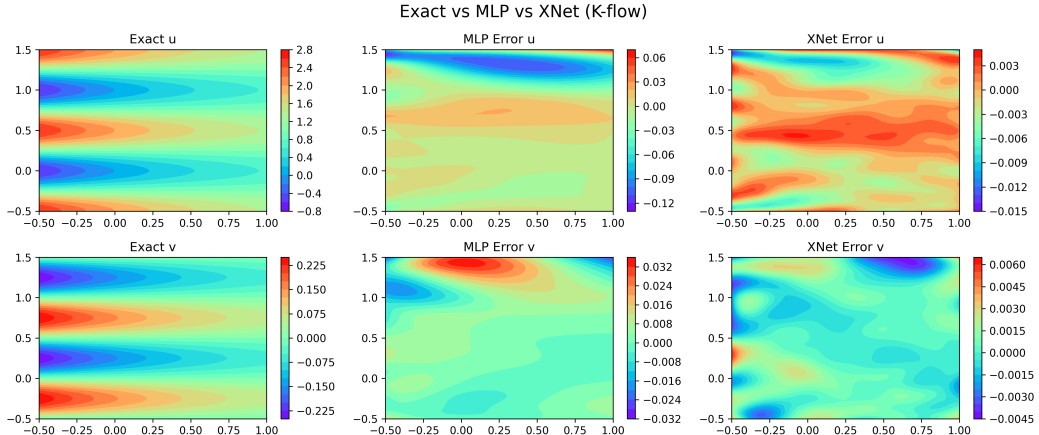

Figure 6: **Kovasznay (Appendix).** Velocity fields and error maps for PINN vs. CauchyNet. Same grid/optimizer/budget as Table 4. Color scales are matched per component.

## B  USE OF LARGE LANGUAGE MODELS (LLMs)

LLMs were used in a limited capacity during the preparation of this manuscript, primarily for:

**Writing assistance:** Grammar checking, sentence restructuring, and formatting consistency across sections. LLMs helped improve clarity of mathematical exposition and experimental descriptions.

**Literature review support:** Assistance in identifying relevant citations and summarizing background work on rational approximation and neural network architectures.

**Code documentation:** Help with commenting and documenting implementation details for reproducibility.

**No contribution to core research:** LLMs did not contribute to the mathematical derivations, theoretical proofs, experimental design, or interpretation of results. The core intellectual contributions - the development of CauchyNet architecture, theoretical rate analysis, and experimental validation - were conducted entirely by the human authors.

**Responsibility:** The authors take full responsibility for all mathematical content, experimental results, and scientific claims in this paper. All technical content has been verified independently of any LLM assistance.

