# OpenReview forum: "Neural Implementations of Rational Approximation: CauchyNet and XNet"
_ICLR.cc/2026/Conference — ICLR 2026 Conference Withdrawn Submission_

### Official Review · Reviewer_NnQh · 2025-10-17

**Soundness:** 1
**Presentation:** 1
**Contribution:** 1
**Rating:** 2
**Confidence:** 4

**Summary:**

The paper introduces CauchyNet, a neural network architecture based on multivariate partial fractions. After introducing the architecture, the authors apply it within the context of physics-informed neural networks and demonstrate its effectiveness on several benchmark problems.

**Strengths:**

- The idea of using rational activation functions has been proposed in the literature and is motivated by the gain of expressivity. The authors build on this idea and propose a novel architecture based on multivariate partial fractions.
- The comparison of the authors between their architecture and a PINN with a fixed activation function shows promising results in terms of final accuracy.

**Weaknesses:**

- Section 2 is very hard to follow since the architecture is mostly defined in one dimension and the definition through the tensor product to high dimensions is not very well explained.
- After simplification in Section 2.3, it seems that the architecture boils down to using a low degree adaptive rational activation function in each neuron. If this is the case, the novelty of the architecture is limited over Boulle et al. and Molina et al.
- The authors mention that existing rational neural networks lack connection to approximation theory. I strongly disagree with this statement, the approximation rates from Boulle et al. are quasi optimal (to approximate functions with $k$ derivatives in dimension $d$ and Telgarsky also provides approximation results for rational neural networks). One can simply extend these results to analytic functions to show exponential convergence.
- The authors mention that one of their main contributions is to prove convergence rate for their architecture. However, there is no theory in the paper besides the citation of the known rational approximation result from Newman and Stahl. It is not clear that the proposed architecture achieve this rate given the softplus regularization.
- There is a lack of core references from the literature on the use of different activation functions (including adaptive ones in PINNs by Jagtap et al.), rational functions and neural networks (e.g. by Telgarsky), and approximation theory for neural networks (e.g. by Yarotsky).

**Questions:**

- Could the authors report the computational cost of training CauchyNet compared to a standard PINN?
- Could the authors clarify the novelty of their architecture compared to existing rational neural networks with adaptive activation functions?
- The numerical experiments should be compared with neural networks using adaptive activation functions (see Jagtap et al.) and previous rational neural networks architectures. At the moment, the proposed architecture is larger than the baselines due to the rational weights.

---

### Official Review · Reviewer_WpTv · 2025-10-30

**Soundness:** 3
**Presentation:** 2
**Contribution:** 1
**Rating:** 2
**Confidence:** 4

**Summary:**

This paper presents two neural architectures, CauchyNet and XNet, which are based on the principle of replacing standard activation functions with learnable rational approximations. CauchyNet is presented as a direct realization of the multivariate Cauchy integral formula, summing a series of atoms. The XNet is a more efficient implementation which uses ridge-projections of 1D Cauchy atoms to scale to high-dimensional applications. The authors claim these methods preserve exponential or root-exponential approximation rates for analytic and piecewise-analytic functions, and show several experiments including function approximation, solving PDEs, and generative modeling.

**Strengths:**

- The authors provide code. I have run this and the results are consistent with those reported in the paper.
- The authors test across several domains and demonstrate that the Cauchy/XNet layers can be trained stably with good results.
- The idea of integrating rational function theory into neural architectures is conceptually appealing and might inspire new directions in neural approximation theory.

**Weaknesses:**

1. Limited Novelty. The main activation, as well as much of the detail on the CauchyNet and XNet, has already been introduced in "Cauchy activation function and XNet," reference [6] in the original work. This paper appears to just implement these within a neural network. While the connection to classical rational approximation is clearly emphasized, it is hard to say that this constitutes a novel contribution beyond the prior work in this field.
2. Weak experimental baselines. The baselines are limited to shallow MLPs with standard activation functions. PINNs have seen many improvements from the naive implementation. Moreover, there are many PDEs which exhibit "sharp local features" which the authors suggest would be modeled more accurately with rational approximation. Demonstrating a drop-in replacement of an MLP by a CauchyNet in a challenging problem where even modern alternatives to PINNs (FBPINNs, PIKANs) would be more convincing. Likewise, the RationalNet (ref. 1 in the original work) or Kolmogorov-Arnold networks [arXiv 2404.19756] also use learnable activations (whether rational or otherwise) and see little-to-no comparison in this work.
3. Limited literature context. Related to the previous point, RationalNet and KANs are not only two examples of alternative baselines, but also serve as two examples of related works which are not adequately discussed in this work. I encourage the authors to review related literature and include a broader set of sources, as well as sufficient descriptions of their work in this broader context.
4. Scalability not convincingly demonstrated. Although XNet is claimed to scale linearly, there are no experiments which demonstrate this. I attempted to reproduce large-scale tests of an autoencoder with XNet or CauchyNet replacements for some layers. I found a lot of difficulty in scaling these up to the order of millions of parameters and maintaining stable gradients. I used several tricks from the paper to ensure stability, but in the end I still found that the model did not do well to model functions with sharp gradients and detailed features. The paper would benefit from a true large-scale implementation to substantiate the "scalable" claim.

**Questions:**

1. How does this work differ conceptually and technically from "Cauchy activation function and XNet"? Are CauchyNet and XNet an extension or merely an implementation of that prior architecture?
2. Can the authors provide results on more competitive baselines (mentioned above)?
3. Could the authors provide experiments which demonstrate the ability to scale to realistic scenarios, beyond toy problems?
4. How sensitive is the approach to the choice of the denominator offset? Is there any analysis on the stability landscape or conditioning for varying this hyperparameter?
5. Can the authors comment in technical detail on how their proposed approaches compare to rational activation functions with trainable poles? For example, PAU [arXiv 1907.06732] and rational ReLU [arXiv 2502.06283].
6. Is it possible to ensure that the pole-separation margin remains controlled during training deep networks, beyond the preservation rates presented in the paper? Would this introduce something like an implicit regularization constraint, and would this have the potential to limit expressivity at the cost of stability?

---

### Official Review · Reviewer_48b8 · 2025-10-30

**Soundness:** 3
**Presentation:** 3
**Contribution:** 1
**Rating:** 6
**Confidence:** 4

**Summary:**

The authors aim to construct an architecture that inherits the optimal convergence rates of rational approximants while avoiding the curse of dimensionality which follows from tensor product constructions to reach high dimensions. They do this by using XNets to construct a modular layer that builds ridge-projected Cauchy layers. In 1D the method reduces to the classical theory, and empirical measurement are used to gauge accuracy/capacity in higher dimensions.

This is a mathematically sound and nice paper with a clear tie to classical approximation theory. The authors demonstrate substantial accuracy gains for a variety of benchmarks.

Overall this is nice but straightforward. The results outperform MLPs for simple regression and PINN tasks. It would have been nice to see some more challenging comparisons (e.g. to deep resnets or transformers).

**Strengths:**

The problem is well motivated and provides clear advantages over mlps.

**Weaknesses:**

The evidence is primarily experimental, and improvement is demonstrated on simple problems against weak competiting architectures. Additional comparisons to more challenging benchmarks might strengthen the impact, otherwise this strikes me as a "make an applied math idea trainable and see if it does good regression" paper, which I personally find interesting but may be of less interest to the ICLR community.

**Questions:**

no questions

---

### Official Review · Reviewer_WAEZ · 2025-11-03

**Soundness:** 2
**Presentation:** 3
**Contribution:** 2
**Rating:** 6
**Confidence:** 3

**Summary:**

This paper introduces CauchyNet and XNet, which are model architectures that are similar to rational function interpolants, based on the Cauchy integral formula. The architecture is obtained by discretizing the integral formula using quadratures. In one dimension, this inherits theoretical rates from classical rational approximant theory. Directly scaling the quadrature with dimension would result in exponentially many poles, so the paper proposes an alternative high dimensional model called XNet. This is obtained by projecting the higher dimensional Cauchy layer along a few ridge directions, leading to linear scaling with dimension. Numerical experiments showcase improved performance of these architectures compared to baselines in common PDE problems.

**Strengths:**

* The paper is well written and easy to follow.
* Theory seems to be sound and relevant.
* The architecture seems to be novel, at least in the higher dimensional case (XNet).
* Numerical results are strong and convincing.

**Weaknesses:**

* It seems like the theory for the high dimensional model is a bit narrow, in that in needs separability of the ridges.
* Improvements over classical rational and spectral methods is a bit unclear, and the gap to prior work is also not that clear to me.
* The main benefit of the ridge-projected CauchyNet is the dimension scaling. However, as per my understanding, all experiments are done in 3 dimensions or less.

**Questions:**

1. What is the scope for theory in non-ridge settings?
2. Could the authors elaborate on the improvement of this approach to classical spectral methods for solving PDEs?

---

### Note · Authors · 2025-11-20

I have read and agree with the venue's withdrawal policy on behalf of myself and my co-authors.